

# RoadSurf 1.1: open-source road weather model library

**Virve Karsisto**[1]

[1]Finnish Meteorological Institute, Helsinki, Finland

**Correspondence:** Virve Karsisto (virve.karsisto@fmi.fi)

**Abstract.** Icy and snowy road conditions cause problems in many countries where temperature often drops below zero degrees. Preventive actions are necessary to keep roads ice-free and to optimize maintenance operations. Accurate road surface temperature and road condition forecasts are an essential help for road maintenance crews to plan their actions. Finnish Meteorological institute (FMI) has produced road weather forecasts for many years with the house-made road weather model RoadSurf to aid local road authorities. Recently, FMI published an open-source road weather model library that consists of RoadSurf functions. The open publication provides an opportunity for many institutes and companies to use the library in road weather forecasting. The evaluation of the library shows that it is well suited for forecasting road surface temperature.

## 1 Introduction

RoadSurf is an open-source library for predicting winter road conditions. It is published under MIT license and contains code from the Finnish Meteorological Institute's Road Weather model (Kangas et al., 2015), that is also named RoadSurf. In principle, it is the original RoadSurf model refactored into the library format. In this work, the model physics and algorithms are explained in detail. A model description paper of the original RoadSurf has been provided by Kangas et al. (2015), but the model has been further developed since then. Therefore, an updated model description is necessary. The newest version of the RoadSurf library is available in FMI developers GitHub page (https://github.com/fmidev/RoadSurf) and the version used in this study is available at Zenodo (Karsisto et al., 2023). Functions in the library can be used to forecast road surface temperature and the amount of water, ice, snow, and black ice on the road. Accurate predictions of road surface temperature and road conditions are important for ensuring traffic safety, minimizing the economic impacts of hazardous weather events, and maintaining efficient traffic flow. For example, with accurate surface temperature forecasts, road maintenance crews can be prepared for situations where the temperature drops below zero and salt the roads before ice starts to form. In addition, unnecessary salting can be avoided, resulting in economic savings. Accurate forecasts are also important to transportation and logistics, as the forecasts allow companies to optimize transportation routes and prepare for eventual disruptions to supply chains.

Several different road weather models have been developed around the world (Rayer, 1987; Yang et al., 2012; Karsisto et al., 2017). Perhaps the most used one is the open-source METRo model developed in Canada (Crevier and Delage, 2001). The development of the RoadSurf model started in the late 1990s and its operational use started in 2000. Since then, the model has been under continuous development. For example, friction calculation and shadowing algorithm have been added to the model (Juga et al., 2013; Karsisto and Horttanainen, 2023). The model can also be used to forecast sidewalk conditions (Hippi et al., 2020). The RoadSurf library contains the physical functions used by the RoadSurf model to predict road conditions. It does not contain determination of all variables predicted by RoadSurf model, such as friction and road condition (wet, icy, snowy etc.). However, the library is designed so that each user can make their own implementation. While the RoadSurf library provides the basic prediction functions, each user can make their own additions to the model simulation. The RoadSurf library Zenodo archive provides an example implementation, which can be used as a starting point.





The functions in the RoadSurf library model the ground temperature profile and surface energy balance in one dimension. Information about atmospheric variables, such as air temperature and wind speed, is required to make the simulation. These can be obtained from the forecasts made by numerical weather prediction (NWP) models. The workings of RoadSurf model have already been presented in several publications (Kangas et al., 2015; Karsisto et al., 2016, 2017; Karsisto and Horttanainen, 2023), but this article provides an update on the new open-source RoadSurf library. In addition, a small evaluation study is included to assess the library's performance. More extensive evaluations of the original model are provided by e.g. Karsisto et al. (2016, 2017); Toivonen et al. (2019); Hippi et al. (2020); Karsisto and Horttanainen (2023).

The RoadSurf library can be used anywhere with the condition that weather forecast data are available. The RoadSurf model has been run for several countries in addition to Finland, including, for example, Norway, Sweden, Estonia, and The Netherlands. When implementing the model to a new area, it is important to adjust the model variables such as asphalt heat capacity and density to fit to the local conditions. The model is not optimized to predict very high road surface temperatures, so it should not be used to predict damage caused to the road by excessive heat without further studies.

## 2   Requirements

RoadSurf library is coded in Fortran and tested with GNU Fortran (GCC) 8.5.0. The library is not guaranteed to work with older Fortran versions. As input data, the RoadSurf library requires the following variables:

– Air temperature (°C)

– Dew point temperature (°C) or Humidity (%)

– Wind speed ($ms^{-1}$)

– Precipitation ($mmh^{-1}$)

– Incoming long-wave radiation ($Wm^{-2}$)

– Incoming short-wave radiation ($Wm^{-2}$)

Optional values are road surface temperature (°C) and precipitation phase. In case sky view factors and local horizon angles are used to modify the incoming radiation, the model also requires surface net long-wave radiation ($Wm^{-2}$) and direct solar radiation ($Wm^{-2}$) as input values. Sky view factor describes the fraction of the radiation flux reaching the surface versus the radiation from the whole hemisphere. Local horizon angle means the angle between the visible horizon and the ground in a certain direction.

The atmospheric variables should form a time series for the simulation duration. Although the model simulation can be done by using only either observation or forecast data, it is intended that the input data for the first part of the simulation consist of observations. This way, the ground temperature profile can evolve and form a good starting state for the actual forecast. The forecast data are necessary for forecasting the evolution of the temperature profile. If road surface temperature observations are available, they can be used to force the simulated surface temperature to the observations during the initialization.

As the RoadSurf library contains only functions used by the RoadSurf model, the user is responsible for creating a program that implements them. Tow examples of such program is given in the RoadSurf library GitHub repository. The user can write their own functions to read the input data or even include the calculation of additional variables to the simulation without modifying the original library.

## 3   RoadSurf physics

This section gives detailed explanations of the RoadSurf library physics. The notations used in this section are summarized in appendix A for easier reference.

### 3.1   Surface energy balance

The most important equation in the model is the surface heat balance equation, which manages different energy flows in and out of the surface (Brutsaert, 1984):

$$G = R_n - LE + H + Tr, \tag{1}$$





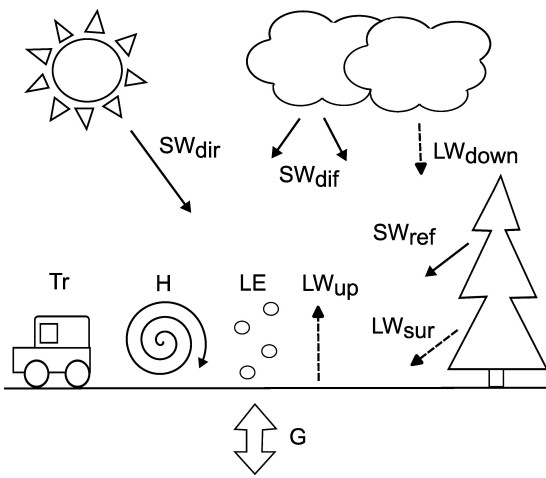

**Figure 1.** Schematic representation of model fluxes.

where

| | | |
|---|---|---|
| $G$ | = | Ground heat flux, |
| $R_n$ | = | Net radiation , |
| $LE$ | = | Latent heat flux , |
| $H$ | = | Sensible heat flux , |
| $Tr$ | = | Heating caused by traffic. |

Heating caused by traffic is just a constant value that can take on different values during daytime and nighttime. The following sections describe the calculation of the other fluxes in detail. Figure 1 shows a schematic representation of the energy fluxes.

### 3.1.1 Net radiation

Net radiation consists of incoming long-wave and short-wave radiation, black body emission from the asphalt and reflected short-wave radiation (Brutsaert, 1984):

$$R_n = SW_{down}(1 - \alpha_s) + LW_{down} - LW_{up}, \tag{2}$$

where

| | | |
|---|---|---|
| $SW_{down}$ | = | Incoming short-wave radiation, |
| $\alpha_s$ | = | Surface albedo, |
| $LW_{down}$ | = | Incoming long-wave radiation, |
| $LW_{up}$ | = | Black body emission. |

If sky view factor and local horizon angels are given, they are used to modify radiation fluxes. Surrounding buildings, terrain and vegetation reduces the amounts of incoming diffuse short-wave radiation and incoming long-wave radiation from the sky. However, they also emit some long waver radiation and reflect short wave radiation. Sky view factor is used to take these effect into account. In addition, local horizon angles are used to determine whether the forecast point receive direct solar radiation. (Karsisto and Horttanainen, 2023).

The input radiation values are from a numerical weather prediction model and describe values for the whole grid cell. Therefore, the net long wave radiation or reflected solar radiation do not describe well the values for a single road point. However, when sky view factor is used, also the radiation from surrounding buildings, vegetation and terrain is taken into account. The values from the numerical prediction model can be used to estimate the reflected/emitted radiation from the surrounding environment.





Diffuse solar radiation is not included in the input variables, so the amount of diffuse solar radiation is calculated as the difference between total downward short wave radiation flux and the direct radiation flux:

$$SW_{dif} = SW_{down} - SW_{dir} \tag{3}$$

Sky view factor is used to reduce the diffuse solar radiation from the sky and to take into account the reflected solar radiation from the surrounding (Senkova et al., 2007):                                                                      5

$$SW_{dif,mod} = SVF * SW_{dif} + (1 - SVF)SW_{ref} \tag{4}$$

where $SW_{ref}$ is the reflected solar radiation from surroundings. It is calculated as sum of reflected direct solar radiation and reflected diffuse solar radiation fluxes:

$$SW_{ref} = \alpha_{sur}SW_{dir} + \alpha_{sur}SW_{dif} \tag{5}$$

where $\alpha_{sur}$ is the albedo of the surrounding environment. The modified incoming solar radiation is then calculated as:        10

$$SW_{down} = SW_{dif,mod} + SW_{dir} \tag{6}$$

If local horizon angles are given, the model determines whether the point gets direct solar radiation based on the sun position. The sun position calculation is based on book by Jean Meeus Meeus (1991). If the local horizon angle in the direction of the sun is greater than the sun elevation angle, the direct solar radiation is set to zero.

Sky view factor is used to modify the incoming long wave radiation as (Senkova et al., 2007):                                         15

$$LW_{down} = SVF * LW_{down} - (1 - SVF)LW_{sur} \tag{7}$$

where $LW_{sur}$ is long wave radiation from the surrounding vegetation, terrain and buildings. The upward radiation from from surroundings can be calculated as the difference of the net long wave radiation and incoming long wave radiation given as input to the model:

$$LW_{sur} = LW_{net} - LW_{down} \tag{8}$$        20

This value is used to give an approximation of the radiation from the surroundings towards the road point. The long wave radiation emitted by the road is calculated as black body emission (Campbell, 1986):

$$LW_{up} = \epsilon\sigma_{SB}T_s^4, \tag{9}$$

where
$\epsilon$ = emissivity factor ,
$\sigma_{SB}$ = Stefan-Boltzmann constant,        25
$T_s$ = surface temperature.

### 3.1.2 Albedo

Albedo means the fraction of the shortwave radiation that is reflected by the surface. Both asphalt albedo snow albedo are given to RoadSurf as input parameters. Default values are 0.1 and 0.6, respectively. Asphalt albedo is used when there is no ice, snow or deposit on the road surface. When there is snow on the surface, the alpedo is set to snow albedo as long as the snow storage exceeds the ice storage. When the ice storage is larger than the snow storage, the alpedo is calucalted with equation:        30

$$\alpha_s = \alpha_{asp} + \frac{St_{sum,ice}/1.5}{\alpha_{snow} - \alpha_{asp}}, \tag{10}$$

where $\alpha_{asp}$ is asphalt albedo, $\alpha_{snow}$ is snow albedo and $St_{sum,ice}$ is the total ice content. The total ice content is calculated as the sum of deposit storage and the average of the ice storage and the secondary ice storage. Secondary ice storage is explained in section 3.5. If $St_{sum,ice}$ is greater than 1.5 mm, the albedo is set to snow albedo.





### 3.1.3 Sensible heat flux

Sensible heat flux means the energy flux between air and surface. It is calculated as (Campbell, 1985):

$$H = BLC(T_s - T_a),\tag{11}$$

where $BLC$ is boundary layer conductance. Boundary layer conductance describes how easily the heat is transferred between air and surface. It is calculated as (Campbell, 1985, 1986):

$$BLC = \frac{c_a \rho_a k u^*}{ln(\frac{z_T - d + z_h}{z_h}) + \Psi_h},\tag{12}$$

where
$c_a$ = specific heat of air,
$\rho_a$ = density of air,
$k$ = von Karman's constant,
$u^*$ = friction velocity,
$z_T$ = Air temperature measurement height,
$d$ = zero plane displacement height,
$z_h$ = surface roughness parameter for heat,
$\Psi_h$ = stability correction factor for heat.
An iterative process is applied to solve this equation as it cannot be solved directly because of connected variables.
$c_a$ is dependent on air temperature and is calculated as isobaric specific heat for dry air (Garratt, 1992):

$$c_a = 1005 + \frac{(T_a - 250)^2}{3364},\tag{13}$$

where $T_a$ is air temperature. Air density is calculated according to the ideal gas law:

$$\rho_a = \frac{p}{R_d T_a},\tag{14}$$

where $p$ is pressure and $R_d$ is gas constant for dry air (287.05 $JKg^{-1}K^{-1}$). Pressure is not included in the input of the simulation and is assumed to always be 1000 hPa.
$u^*$ is calculated with equation Campbell (1985):

$$u^* = \frac{ku}{ln(\frac{z_W - d + z_m}{z_m}) + \Psi_m},\tag{15}$$

where $z_W$ is wind speed measurement height, $z_m$ is surface roughness parameter for momentum and $\Psi_m$ is stability correction factor for momentum. Values for $z_T$, $z_W$ $d$, $z_h$, and $z_m$ are given as input. Default values are $z_T$=2 m, $z_W$=10 m , $d$ =0 and $z_h$=0.001 m, $z_m$=0.4 m.

Boundary layer stability is an important factor in the sensible heat flux calculation. Mixing is larger in unstable conditions than in a stable boundary layer. Relative importance of the thermal and mechanical turbulence in boundary layer transport is described as (Campbell, 1985):

$$\zeta = -\frac{k z_T g H}{c_a \rho_a T_a u^{*3}},\tag{16}$$

where $g$ is gravitational constant (9.81 $ms^{-2}$). $\zeta$ is used to calculate stability correction factors. In stable conditions $\zeta$ is positive and

$$\Psi_h = \Psi_m = 4.7\zeta.\tag{17}$$





In unstable conditions $\zeta$ is negative and

$$\Psi_h = -2ln(\frac{1 + \sqrt[2]{1 - 16\zeta}}{2}) \tag{18}$$

and

$$\Psi_m = 0.6\Psi_h. \tag{19}$$

The equations above include several connected parameters which is why $BLC$ must be solved iteratively. First, $\Psi_h$ and $\Psi_m$ 5
are set to zero and $BLC$, $H$ and $\zeta$ are calculated. Second, $\Psi_h$ and $\Psi_m$ are calculated using the obtained values. Then $BLC$,
$H$ and $\zeta$ are calculated again using the new values. The iteration is continued until the absolute difference between two $BLC$
values from the subsequent iteration rounds is smaller than 0.001 or the maximum amount of iterations (40) is reached.

### 3.1.4  Latent Heat Flux

Latent heat flux means the energy released in condensation or energy consumed in evaporation. It is calculated as (Calder, 10
1990):

$$LE = \frac{\rho_m c_a}{\gamma} \frac{e_s - e_a}{r_o} \tag{20}$$

where
| | | |
|---|---|---|
| $\rho_m$ | = | density of moist air, |
| $\gamma$ | = | psychrometric constant, |
| $e_s$ | = | vapour pressure at the surface, |
| $e_a$ | = | vapour pressure of the air |
| $r_o$ | = | aerodynamic resistance. |

Positive latent heat flux means that water is evaporated from the surface and negative that water is condensed on the surface. 15
$LE$ is set to zero if there is no water to evaporate and $LE$ is positive. In the calculation $\rho_m$ is approximated as $\rho_a$. $\gamma$ is dependent
on air temperature and is calculated using equation developed from values in Calder (1990):

$$\gamma = 0.1 * (0.00063T_a + 0.47496), \tag{21}$$

Water vapour pressure at the surface is approximated as saturated water vapour pressure. Over water, this can be calculated
as (Calder, 1990): 20

$$e_s = 0.61078e^{17.269T_s/(T_s + 237.3)} \tag{22}$$

and over ice as

$$e_s = 0.61078e^{21.875T_s/(T_s + 265.5)} \tag{23}$$

In the air, the water vapor pressure is calculated from the saturated vapor pressure as:

$$e_a = \frac{RH}{100} * e_s, \tag{24}$$ 25

where RH is relative humidity. In unstable conditions, aerodynamic resistance can be calculated as (Tourula and Heikinheimo,
1998):

$$r_o = \frac{(ln(\frac{h-d}{z_m}) + \Psi)(ln(\frac{h-d}{z_h}) + \Psi)}{k^2 u}, \tag{25}$$





where $h$ is measurement height and $\Psi = \Psi_m = \Psi_h$. However, RoadSurf library uses modified version of this equation:

$$r_o = \frac{(ln(\frac{z_W+z_m}{z_m}) + \Psi_m)(ln(\frac{z_W+z_h}{z_h}) + \Psi_h)}{k^2 u}. \tag{26}$$

$\Psi_m$ and $\Psi_h$ are not considered to be equal as in Tourula and Heikinheimo (1998), where stability correction was marked simply as $\Psi$. In addition, zero plane displacement is assumed to be small, but surface roughness for momentum is added to the equation. The aerodynamic resistance has maximum limit of 30 $sm^{-1}$. The difference between equations 25 and 26 cause only slight differences in aerodynamic resistance, so the modifications do not have much effect. Equation 26 is used also in stable conditions in the simulation for simplicity.

### 3.2 Heat flow in the ground

The ground is divided into several layers for the heat flow calculation. The number of layers is 16. The heat transfer between the layers is based on equation Patankar (1980):

$$\rho_g c_g \frac{\partial T(z,t)}{\partial t} = \frac{\partial}{\partial z} K \frac{\partial T(z,t)}{\partial z}, \tag{27}$$

where
| | | |
|---|---|---|
| $T$ | = | layer temperature, |
| $z$ | = | vertical distance in the ground, |
| $t$ | = | time, |
| $K$ | = | heat conductivity, |
| $\rho_g$ | = | density, |
| $c_g$ | = | specific heat capacity of the ground. |

After integrating over the volume of the layer and time step, the equation is discretized and solved with forward difference explicit method. This results in the following equation Campbell (1985):

$$T_i^{j+1} = T_i + \frac{1}{\rho_g c_g \frac{z_{i+1}-z_{i-1}}{2\Delta t}}(K_i \frac{T_{i+1}^j - T_i^j}{z_{i+1} - z_i} - K_{i-1} \frac{T_i^j - T_{i-1}^j}{z_i - z_{i-1}}). \tag{28}$$

The index $j$ refers to the time step and index $i$ to the number of the ground layer. This equation is used to calculate temperature for each layer at each time step. The output surface temperature given by the model is calculated as the average temperature of the first two layers, because it matches better to the observations than the temperature of the topmost layer. The user can also set the height for which the output temperature is interpolated. The average surface temperature is often used in library functions for heat flow and storage term calculation instead of the temperature of the topmost layer.

#### 3.2.1 Volumetric heat capacity and heat conductivity

Water in the ground is taken into account in the heat capacity calculation. The volumetric heat capacity of the ground ($\rho_g c_g$) is calculated as as a weighted average of the dry ground and water volumetric heat capacities:

$$\rho_g c_g = (1 - \phi)\rho_s c_s + \phi \rho_w c_w, \tag{29}$$

where
| | | |
|---|---|---|
| $\rho_g c_g$ | = | volumetric heat capacity of wet ground , |
| $\rho_s c_s$ | = | volumetric heat capacity of dry ground , |
| $\rho_w c_w$ | = | volumetric heat capacity of water , |
| $\phi$ | = | porosity of the ground . |

Porosity means the fraction of empty spaces in the ground. The values are given as input to the library.

Water density and specific heat capacity are considered temperature dependent when temperature is above 0 °C. They are calculated with equations that are derived from tables presented in Campbell (1986) and Weast (1975):





$$\rho_w(T_w) = -0.0050T_w^2 + 0.0079T_w + 1000.0028 \tag{30}$$

$$c_w(T_w) = 1.02 \cdot 10^{-5}T_w^4 - 1.7169 \cdot 10^{-3}T_w^3 + 0.11516T_w^2 - 3.4739T_w + 4217.2, \tag{31}$$

where
$\rho_w$ = water density,
$c_w$ = specific heat capacity of water,
$T_w$ = water temperature.

Water density and specific heat capacity have constant values below zero degrees (Oke, 1987):
$\rho_w$ = $920\ kgm^{-3}$,
$c_w$ = $2100\ kJkg^{-1}K^{-1}$.

Heat conductivity in the ground is calculated as (Campbell, 1985):

$$\lambda = A + B\theta - (A - D)e^{-(C\theta)^E}, \tag{32}$$

where
$\theta$ = volumetric water content .

Coefficients A-E depend on the density of the ground (Campbell, 1985):

$$A = 0.65 - 0.78\rho_g + 0.60\rho_g^2 \tag{33}$$
$$B = 1.06\rho_g \tag{34}$$
$$C = 1 + (2.6/\sqrt{(m_c)}) \tag{35}$$
$$E = 4 \ , \tag{36}$$

where
$m_c$ = clay fraction of the ground .

### 3.2.2 Temperature field and layer height initialization

The ground is divided to 16 layers in the model. The layer thickness grows towards the bottom according to the formula:

$$Z_{i+1} = Z_i + 0.01442(i - 1) + Z_{Add}, \tag{37}$$
$$Z(1) = 0 \tag{38}$$

where $Z_{i+1}$ is the distance of layer $i$ midpoint from the surface and $Z_{Add}$ is constant. The temperature of the layers 1-4 is set to the observed temperature at the initialization. If observations are not available, the temperature of the layers is set to the air temperature. The temperature of the lowest layer depends on the time of the year according to the equation (Monteith, 1975):

$$T_{m+1} = Tc_{m+1} + A_y sin(\Omega J + \Omega Di - \frac{Z_{m+1}}{d_a}) \tag{39}$$

where $A_y$ is the amplitude of variation during the year, $\Omega$ is frequency of the variation $(2\pi/365)$, $J$ is the Julian day, $Tc$ is the climatological temperature average, $Di$ is the displacement of the curve, m is the number of the ground layers and $d_a$ is damping depth. The example implementations uses $Tc = 6.4$, $A_y = 0.6$, $Di = -170$ and $d_a = 2.7$.

Temperature of the layers between the fourth layer and the bottom layer are initialized so that the temperature change is linear with depth.

### 3.3 Coupling

Coupling method can be used to improve the forecast during the first forecast hours (Crevier and Delage, 2001; Karsisto et al., 2016). It is based on adjusting the amount of incoming radiation in the model. The model simulation consists of two phases.





**Table 1.** Events that increase or decrease different storage terms.

|  | Water | Snow | Ice | Black Ice |
|---|---|---|---|---|
| Snowfall |  | + |  |  |
| Rain | + |  |  |  |
| Condensation/Evaporation | +/- |  |  |  |
| Deposition |  |  |  | + |
| Melting | + | - | - | - |
| Freezing | - |  | + |  |
| Wearing by traffic | - | - | - | - |
| Snow packing to ice |  | - | + |  |
| Black ice transfer to ice |  |  | + | - |

First is the initialization phase, where the model is run with observation data. Radiation observations are often missing, so radiation data are taken from the past forecast data. After the initialization phase there is a forecast phase, where the model is run with forecast data. If road surface temperature observations are available, the temperature of the first two layers is set to the observed temperature during initialization.

Coupling phase in the simulation consists of the three hours before the start of the actual forecast phase. During the coupling phase, the simulated surface temperature is not forced to the observed temperature as in the initialization phase. At the end of the coupling phase, the model checks whether the simulated surface temperature fits to the latest observed surface temperature. If not, the model adjusts a radiation correction coefficient that modifies the incoming radiation and the simulation goes back three hours. If the simulated surface temperature is too cold, the coefficient is increased, and if the simulated surface temperature is

too warm, the coefficient is decreased. The model runs the three-hour period again and compares the new simulated temperature to the observation. This is continued until the simulated surface temperature is within 0.1 °C from the observed temperature. Then, the simulation moves to the actual forecast phase. The radiation correction coefficient is used in the forecast phase so that it gradually approaches one as the forecast advances. The used coefficient is calculated as:

$$C_f(t) = 1.0 + C_R e^{-\frac{t}{t_c}}, \tag{40}$$

where $C_f$ is the correcting coefficient at time t, $C_R$ is the original coefficient, and $t_c = 4h$ is a scaling constant. A more detailed explanation about the algorithm that determines the coupling coefficient can be found in the user manual.

### 3.4   Relaxation

When simulation switches from observation phase to the forecast phase, there can be considerable jumps in air temperature, relative humidity and wind speed values as forecast data differs from observation data. Relaxation makes this transition smoother

by modifying these values in the forecast phase with equation:

$$X(t) = X_F(t) - (X_{FO}(t) - X_O)e^{-\frac{t}{t_c}}, \tag{41}$$

where $X(t)$ is the forced value at time t, $X_{FO}$ is the forecasted value at the time of the last observation and $X_O$ is the last observed value. $t_c$ is same scaling constant as in equation 40. Relaxation can be turned on or off.

### 3.5   Storage terms

Storage terms in the RoadSurf library refer to the amounts of snow, ice, water, and deposit on the road (Kangas et al., 2015). The amounts are calculated in water equivalent millimeters. Deposit means black ice that has formed on the road surface via deposition. Unlike the temperature calculation, where the model uses clear physical definitions, storage term calculation contains more approximations and simplifications. There are two separate storages for ice that are otherwise similar but the secondary ice storage is reduced faster by traffic. The secondary ice storage is included in the simulation output. Water storage

consists partly of the water in the asphalt pores and partly of the water on the surface. Water is considered to be only in the pores until water storage exceeds the maximum porous water content, which is given as an input parameter. The default value is 1.0 mm.





Table 1 shows which events increase or decrease the storage terms. Wearing by traffic reduces all storage terms gradually. Passing vehicles compress snow, and with time snow storage is transferred to ice storage. Black ice is transferred to ice storage when there is snow on the road, as black ice is assumed to exist only in snow-free situations.

### 3.5.1   Precipitation

RoadSurf library uses three types of precipitation: water, sleet and snow. When precipitation type is given as input, drizzle, rain, freezing drizzle, and freezing rain are classified as water. Hail is classified as snow. Water type precipitation increases the water storage and snow type precipitation increases snow storage. If precipitation type is sleet, half of the precipitation goes to the water storage and half to the snow storage.

If precipitation type is not given, RoadSurf uses following equation to determine the type (Koistinen and Saltikoff, 1999):

$$P_{rain} = \frac{1}{1 + e^{P_{exp}}}, \tag{42}$$

where $P_{exp} = 22 - 2.7T_a - 0.2RH$ . If $P_{rain} < 0.3$ precipitation is categorized as snow. When $P_{rain} > 0.7$ precipitation is categorized as water and when $0.3 < P_{rain} < 0.7$ precipitation is categorized as sleet.

The storage terms have maximum limits that they cannot exceed. The default maximum limit for snow is 100 mm, for ice 50 mm, for deposit 2 mm and for surface water 1 mm. Ice and water content are simply limited to the maximum amount, but snow storage is halved if it reaches the maximum amount. If deposit exceeds the maximum limit, excess deposit is turned to water.

### 3.5.2   Wear

Passing vehicles cause water, snow and ice to gradually wear off from the road. RoadSurf has separate wear factors for each storage term that determine how fast they are reduced. The reduced amount in one time step can be calculated as:

$$Wear_x = Wf_x * St_x * \Delta t/3600 \tag{43}$$

where $Wf_x$ is the wear factor for storage x and $St_x$ is storage x. The wear factors for different storages are:

$$
\begin{array}{lcl}
Wf_{snow} & = & 0.45 \\
Wf_{ice} & = & 0.319 \\
Wf_{ice2} & = & 2.552 \\
Wf_{deposit} & = & 1.16 \\
Wf_{water} & = & 0.145
\end{array}
$$

Wearing simply reduces the amount if ice, deposit and water on each time step during the simulation. However, part of the reduced snow storage is transformed to ice. This part is calculated by multiplying the reduced amount of snow by 0.556. Snow wear is greater when the snow storage is less than 0.2mm. In that case, snow wear is multiplied by 3. To avoid wearing to get too small, $Wf_x * St_x$ for the snow, ice and deposit wears have a minimum value of 0.01 mm and $Wf_{water} * St_{water}$ has minimum value of 0.06 mm. Deposit wear has effect only in snow free conditions. If $St_{snow} > 0$, then $Wf_{deposit} = 0$.

Water wear is set to zero if the amount of water is less than one tenth of the maximum porous water content. If the water content is less than 90 % of the porous water content, the water wear is reduced by half.

### 3.5.3   Evaporation and condensation

The amount of evaporated or condensed water in time step is calculated with equation:

$$EV_w = \frac{LE}{E_{m^3}}\Delta t, \tag{44}$$

Where $E_{m^3}$ means the amount of energy needed to evaporate one cubic meter of water. It is calculated as:

$$E_{m^3} = L_{wat}\rho_w, \tag{45}$$

where $L_{wat}$ is latent heat of water vaporization and $\rho_w$ is the density of water. Water is evaporated when $LE$ is positive and condensed when $LE$ is negative. Snow, ice and deposit storages must be zero and surface temperature should be at least 0.25 °C for evaporation or condensation to occur in the simulation. If the water storage is less than the maximum porous water





content, the evaporated amount of water is reduced by pore resistance factor. This factor can be set by the user but is by default 1.0, which means it has now effect.

Deposit storage is increased by deposition. It is calculated similarly as condensation but latent heat of evaporation is replaced by latent heat of sublimation in equation 45. Sublimation is not taken into account in the simulation.

## 3.6 Freezing and melting

Freezing happens in the simulation instantly when the surface temperature goes below the freezing limit. When freezing occurs, whole water storage is added to the ice storage and water storage is set to zero. Freezing does not affect temperature in the simulation and heat released by it is not taken into account.

Melting transforms ice, snow and deposit storages into water storage. Whole deposit storage is simply added to the water storage when the surface temperature goes above the melting limit. The melting of ice and snow is slower in the simulation. The amount of melted snow and ice in one time step is calculated as:

$$Me_{snow} = 1000 \frac{Q_{melt}\Delta t}{L_{melt}\rho_w}, \tag{46}$$

where $Q_{melt}$ is the energy that is available for melting and $L_{melt}$ is latent heat for melting. The available energy for melting is considered the same as the energy that would be released if the surface cooled to he melting temperature. It is calculated as:

$$Q_{melt} = (\rho c)_g \frac{z_2 - z_1}{2\Delta t}(T_s - T_{melt}), \tag{47}$$

where $T_{melt}$ is the melting temperature. After calculating the available energy, temperature of the two uppermost layers is set to the melting temperature + 0.01 °C. The melted amount is removed from the ice/snow storage and added to the water storage. If surface temperature warms up on the next time step, the heat is again used for melting and temperature of the two uppermost layer is again set to melting temperature + 0.01 °C. This continues until the ice and snow storages are melted entirely. If the available energy is more than needed to melt the whole ice or snow storage in one time step, the surface is not cooled to the melting temperature but remains warmer. The energy needed to melt the whole storage is calculated as:

$$Q_{all} = \frac{L_{melt}\rho_w \frac{St_x}{1000}}{\Delta t}, \tag{48}$$

where $St_x$ is snow storage if there is snow on the road and ice storage if there is ice but not snow. The left over energy can be calculated simply as the difference between $Q_{melt}$ and $Q_{all}$. Temperature for the uppermost layer is then calculated as:

$$T_1 = T_{melt} + \frac{Q_{melt} - Q_{all}}{(\rho c)_g \frac{z_2 - z_1}{2\Delta t}}. \tag{49}$$

The temperature of the second layer is still set to $T_{melt}$+0.01 °C.

The melting process is not entirely physically correct as the same energy is used to melt both snow and ice storages and $Q_{all}$ includes only ice or snow storage, but the method can still be used to approximate the melting process.

Snow can also be transferred instantly to water if the water content is high enough compared to the snow content. Water to snow ratio is calculated as:

$$WS_{rat} = St_{water,surf}/(St_{water,surf} + St_{snow}), \tag{50}$$

where $St_{water,surf}$ is the part of water storage that is not located in the asphalt pores. If water to snow ratio is larger than 0.6, snow is transferred to water instantly. Snow and water can also turn instantly to ice in the simulation if the snow is wet. If water to snow ratio is higher than 0.1 and surface temperature goes below freezing limit, both water storage and snow storage and transferred to ice storage.

## 4 Data

### 4.1 Forecast

Operational weather forecasts produced by Finnish Meteorological Institute (FMI) were used as input to the road weather model to generate road weather hindcasts. The operational weather forecast is generated by Smartmet nowcast system (Ylhäisi





et al., 2017). It is a combination of a radar-based nowcast, a high-resolution NWP MetCoOp (Meteorological Cooperation on Operational NWP) nowcast and an operational forecast edited by duty meteorologist. The nowcast range (0...6 h) is updated once an hour, whereas the meteorologist edited forecast is updated twice a day.

The radar-based nowcast is generated with FMI Probabilistic Precipitation Nowcasting system (FMI-PPN, https://github.com/fmidev/fmippn-oper/). It uses pySteps ensemble prediction system (Pulkkinen et al., 2019) to generate ensemble forecast from radar data. The final precipitation nowcast is a weighted average of the ensemble members generated from perturbed radar data and the control forecast generated from non-perturbed data (Ylhäisi et al., 2017). The MetCoOp nowcast model is based on convection-permitting Applications of Research to Operations at Mesoscale (AROME) model (Seity et al., 2011). Its horizontal resolution is 2.5 km and it is updated once an hour. The forecast length is 9 hours. The meteorologists at FMI use the SmartMet workstation editing tool (https://github.com/fmidev/smartmet-workstation) to edit the NWP forecast. The meteorologist can either select the base forecast from several different NWP models (ECMWF, MEPS, GFS) or use a mix of several models. The FMI also has different options of post processed data to use as a base forecast. One is based on model output statistics (MOS) (Ylhäisi et al., 2017; Glahn and Lowry, 1972), and the other blends different models (Fritsch et al., 2000; Woodcock and Engel, 2005). After selecting the base model or models, the meteorologist can edit the forecast based on their experience, for example by making certain areas warmer or cooler. The horizontal resolution of the edited data is about 7.5 km and the forecast length is 10 days.

The above-mentioned data sources are blended together to create a seamless forecast for 10 days, in which the first hours consist of nowcast data and the rest of the meteorologist edited forecast. However, the hindcasts generated in this study used only the first 24 hours of the forecast data. Because the data are in grid format, the forecast data were interpolated to the road weather station points by using bilinear interpolation except for the precipitation phase, which was taken from the nearest point instead.

### 4.2 Road weather observations

Intelligent Traffic Management Finland maintains approximately 400 road weather stations in Finland 2. Their data were used in the initialization of the road weather simulations and in the evaluation of the road surface temperature forecasts. The used variables measured by the stations were air temperature, humidity, wind speed, precipitation, and road surface temperature. Road surface temperature is measured either with asphalt-embedded Vaisala DRS511 or optical Vaisala DST111 (Vaisala, 2021). Some road weather stations have multiple sensors, but only one sensor from each station was used. The observation frequency is usually about 5-10 minutes. The observations went through simple quality check routines before using them in the initialization and verification. First, air temperature, surface temperature, dew point temperature, relative humidity, and wind speed values that remained constant for ten or more consecutive observations were removed. Second, spikes were removed from the air temperature, surface temperature and dew point temperature observations by checking if the difference between two consecutive observations was more than five degrees and whether there were two of those kind of changes within four consecutive observations. Third, the values that were greater than 50 °C or lower than -50 °C were removed from the air temperature, surface temperature and dew point temperature observations. The road weather station observations and forecast data are available in FMI Research data repository METIS (Karsisto, 2023b).

### 4.3 Simulation data structure

To evaluate the performance of the RoadSurf library, it was used to make simulations with historical data. These hindcasts were done for the time period from 1 October 2022 to 31 May 2023. The used program that implemented the RoadSurf functions and did the actual simulations can be found in the GitHub repository as "Example 2" in the examples folder. To get a good starting state for the ground temperature profile, the simulations done with RoadSurf library were first run for 48 hours with observation data. However, incoming long-wave and short-wave radiation were obtained from the latest available forecast during the initialization phase due to the lack of radiation observations. After the initialization, the simulation was run 24 hours with forecast data. The hindcasts starting times were 00 UTC, 06 UTC, 12 UTC and 18 UTC each day. The local time in Finland is UTC + 2 hours in winter and UTC + 3 hours when daylight saving time is in effect. For each hindcast, the latest operational forecast that would have been available at that time was selected as the forecast data. Due to technical issues forecasts were missing for time period from 25 December 2 UTC to 26 December 20 UTC. Hindcasts starting during that period used the last forecast generated before the gap.

### 4.4 Example hindcast

Figure 3 shows an example hindcast done with the RoadSurf library. The road weather station had two surface temperature sensors. Sensor 2 gives somewhat colder measurements than sensor 1. The simulated surface temperature fits the observed ones





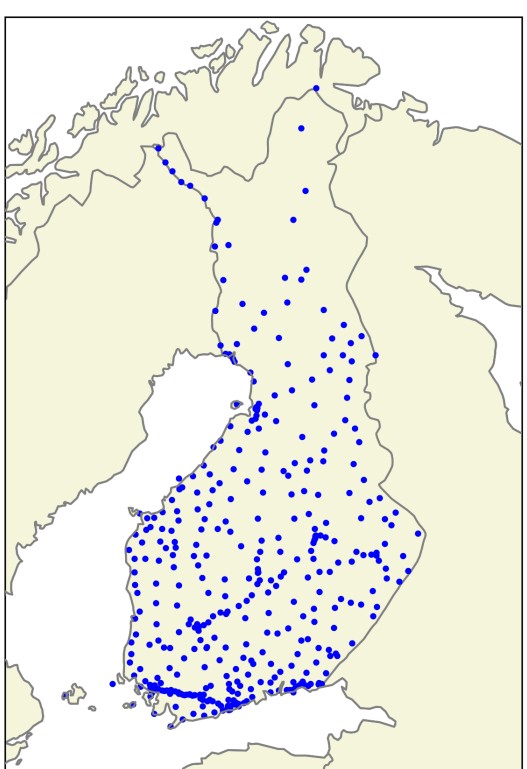

**Figure 2.** Locations of road weather stations. Made with Natural Earth (https://www.naturalearthdata.com/).

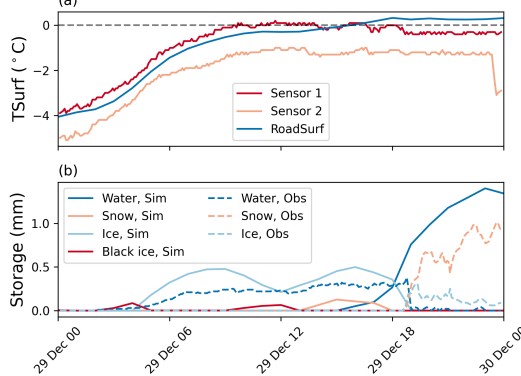

**Figure 3.** An example hindcast made with RoadSurf library. The forecast data start time was 29th December 2022 and the simulation point was Road weather station in Porvoo. The uppermost panel shows simulated (blue line) and observed road surface temperature with two different sensors (red line and peach colored line). The lower panel shows observed (dashed lines) and simulated (continuous lines) amount of water (blue line), snow (peach colored line), and ice (ligth blue line) on the road as water equivalent mm. RoadSurf also simulates black ice, which is shown in red line. The time on x-axis is in UTC time. The colors were chosen with the help of colorbrewer2 (https://colorbrewer2.org).





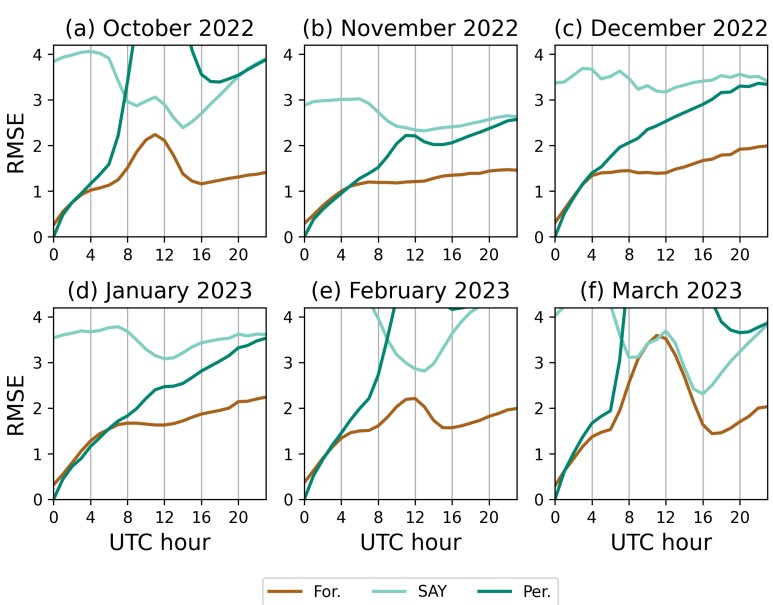

**Figure 4.** RMSE of the road surface temperature forecasts and of the dummy forecasts as a function of UTC time calculated over all road weather stations in Finland. Each panel shows results for one month from October 2022 to March 2023. The forecast start time was 00 UTC each day. Brown lines show the bias of the RoadSurf forecast, light turquoise lines of the dummy forecast forecasting same surface temperature as yesterday at the same time, and the turquoise lines of the dummy forecast forecasting same surface temperature as at the forecast start time. The colors were chosen with the help of colorbrewer2 (https://colorbrewer2.org).

quite well but is somewhat too warm at the end of the simulation. During daytime the surface temperature measured by sensor 1 is close to zero, whereas the simulated surface temperature is slightly colder. The observations show that there is water on the road, whereas in the simulation the road is icier. This turns around in the evening, when the simulated temperature rises below zero degrees but the measured temperature by sensor 1 drops below zero. According to the observations the road gets snowy and icy, but in the simulation the ice melts. This example shows that just a slight difference in predicted and observed temperature can lead to great differences in predicted and observed road condition.

## 5   RoadSurf evaluation

Simulations done with RoadSurf library were evaluated against road surface temperature observations. Figures 4 and 5 show Bias (forecast-observation) and root mean square error (RMSE) for the 00 UTC started hindcasts for each month. These hindcasts are henceforth referred as forecasts for simplicity. The figures also show results for two "dummy" forecasts as a reference. The first dummy forecast always forecasts the same temperature as observed on the previous day at the same time. From now on it will be referred to as SAY (Same As Yesterday). The second dummy forecast is a persistence forecast that always forecast the same road surface temperature as the observation at the forecast start time. The RMSE of the RoadSurf forecast and the persistence forecast are quite similar in the first hours of the forecast. The persistence forecast gives little better results at first, but the error quickly increases to be much greater than for the RoadSurf forecast. The persistence forecast works better in wintertime than in autumn and spring because the daily surface temperature variation is smaller. The SAY forecast gives considerably greater RMSE error than the RoadSurf forecast for all months and lead times except during daytime in March, where they are quite similar. The RoadSurf library has difficulties forecasting maximum temperatures when the daily variation in temperature is large.

The Bias of the RoadSurf forecast is generally quite small. From November to January the forecasts are a little too cold, but in October, February, and March there is some warm bias during daytime. The persistence forecast also has quite small bias during nighttime but has very negative values during daytime. The bias of the SAY forecast is also quite small. In October and





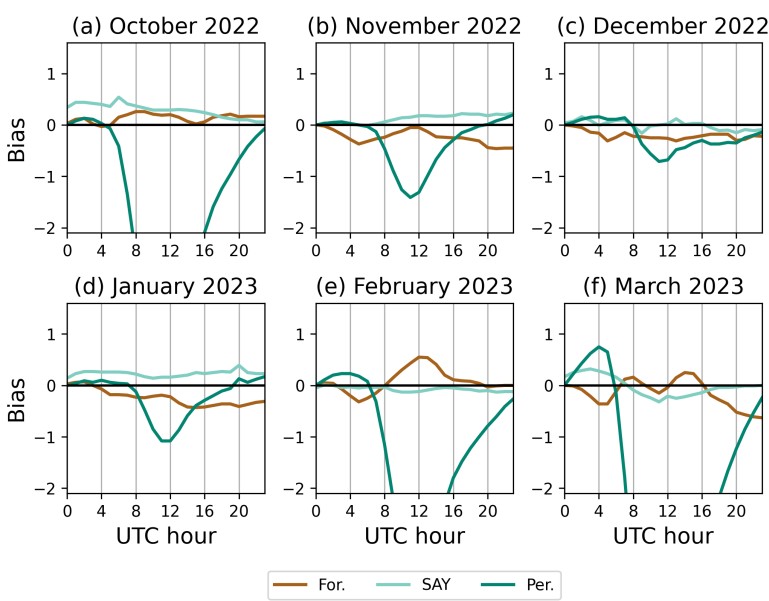

**Figure 5.** Same as Figure 4 but for bias

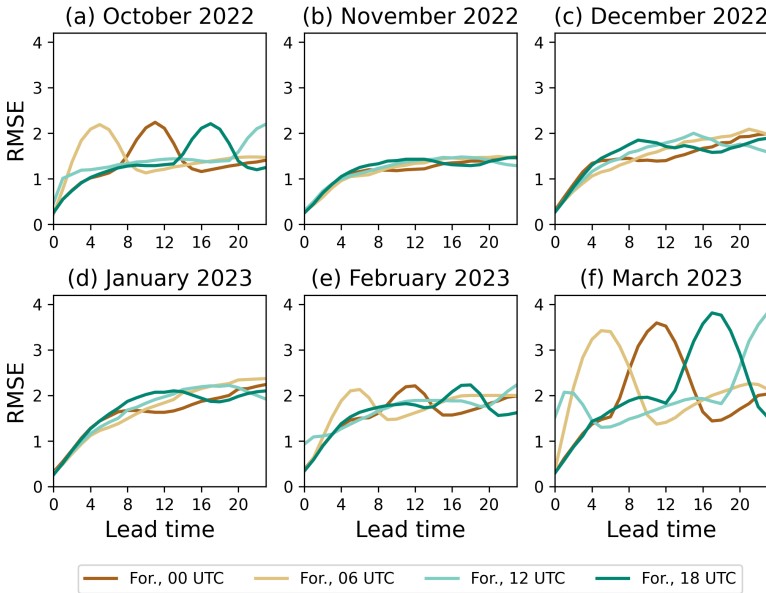

**Figure 6.** RMSE of the road surface temperature forecasts as a function of forecast lead time calculated over all road weather stations in Finland. Each panel show results for one month from October 2022 to March 2023. Brown lines show the bias of the RoadSurf forecast started at 00 UTC, light brown lines of the forecasts started at 06 UTC, light turquoise lines for the forecast started at 12 UTC and turquoise lines for forecast started at 18 UTC. The colors were chosen with the help of colorbrewer2 (https://colorbrewer2.org).



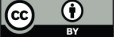

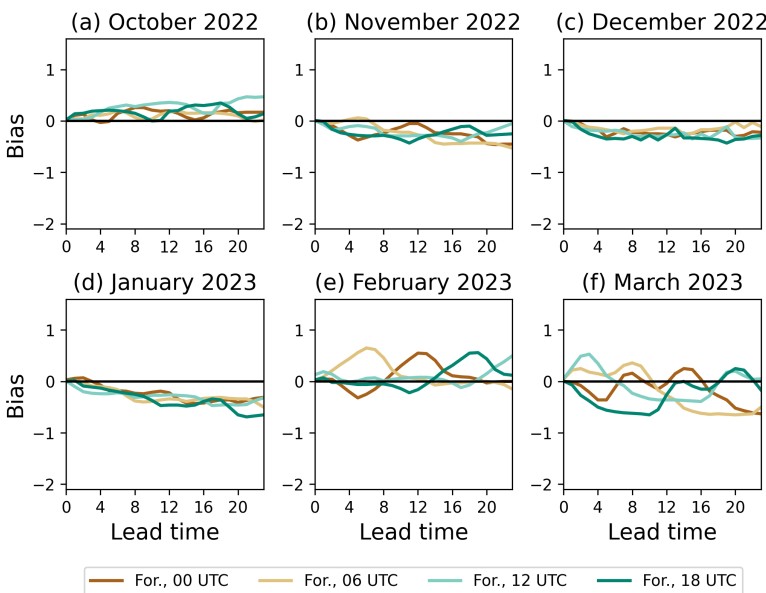

**Figure 7.** Same as Figure 6 but for bias

in January it is generally a little positive, which indicates that there is a general decreasing trend in temperature during those months.

As the daily surface temperature variation is quite large in autumn and spring, the forecasts with different starting times were evaluated separately. Figures 7 and 6 show bias and RMSE for the forecasts started as 00, 06, 12 and 18 UTC. Similarly as 00 UTC started forecasts, the forecasts with other start times have RMSE maxima during daytime in October, February and March. During those months the 12 UTC started forecast has a little higher RMSE at the forecast start time. This is because the forecast starts around the time of the daily surface temperature maximum. From November to January the RMSE values are rather similar, but there is some daily variation as well. The 00, 06 and 18 UTC forecast seem to have smaller RMSE values when the actual forecast time is around 12 UTC.

All forecasts with different starting times have small warm bias in October. During November, December, and January the bias is mostly negative. In February, all forecasts with different start times have some warm bias during daytime. There is also some warm bias in March during daytime but the bias seem to have a local minima around 12 UTC. During nighttime in March all forecast start times have cold bias.

The evaluation results were calculated over all road weather stations in Finland, but naturally the results vary from station to station. The evaluation focused only on road surface temperature because the amounts of water, ice and snow measured by road weather station are not considered reliable enough for proper evaluation. It should be noted that the simulations are very sensitive to the input data. If the atmospheric forecast given as input to the simulation is of poor quality, then probably so is the the road weather forecast. In this evaluation, the atmospheric forecast data were from from FMI operational weather forecast, which is the best data FMI has to offer. The newest operational forecast is available as open data in FMI open data service (https://en.ilmatieteenlaitos.fi/open-data).

## 6   Conclusions

RoadSurf is a Fortran library that contains functions for predicting winter road conditions. The requirements to use it, as well as the main physics and evaluation against observations, have been presented. The evaluation showed that RoadSurf library is well suited for predicting road surface temperature. The persistence forecast gives comparable results during the first forecast hours, but after 4...6 hours the simulations done with RoadSurf perform clearly better. The evaluation results were better for





months when there is less daily temperature variation, as forecasting daytime maximum temperatures is challenging when the daily temperature variation is great.

Forecasts created by RoadSurf can assist in keeping roads safe for drivers wherever wintry road conditions occur. The library allows flexible implementation. Each user can create their own program utilizing the library functions, or they can just implement a few functions to their own road weather model. RoadSurf allows users to create their own input and output handling functions, which makes it easier to implement the library to existing systems. It is advised to run the model with historical data before operational use and adjust the model parameters to fit into local conditions. The model results should also be evaluated regularly against observations. RoadSurf provides an alternative to a broadly used METRo road weather model, giving institutes and companies more options when creating and developing their forecasting systems.

*Code and data availability.* The RoadSurf code used in this study is available in Zenodo (Karsisto et al., 2023) under MIT licence. The program used to run the simulation is located in the same archive under examples folder as "Example2". The newest version of the code is available in GitHub (https://github.com/fmidev/RoadSurf/). Road weather station observations, atmospheric forecast data, simulation results and verification scores are available inf FMI data repository METIS (Karsisto, 2023b). Other data processing and plotting scripts are available in Zenodo (Karsisto, 2023a).

## Appendix A

All notations used in section 3 are explained in Tables A1 - A6 for easier reference.



**Table A1.** Variables describing energy fluxes.

| Notation | Meaning |
|---|---|
| $G$ | Ground heat flux |
| $R_n$ | Net radiation |
| $LE$ | Latent heat flux |
| $H$ | Sensible heat flux |
| $Tr$ | Heating caused by traffic |
| $SW_{dif}$ | Diffuse solar radiation |
| $SW_{dif,mod}$ | Modified diffuse solar radiation |
| $SW_{dir}$ | Direct solar radiation |
| $SW_{down}$ | Incoming short-wave radiation |
| $SW_{ref}$ | Reflected solar radiation from the surroundings |
| $LW_{down}$ | Incoming long-wave radiation |
| $LW_{net}$ | Net long-wave radiation |
| $LW_{sur}$ | Long-wave radiation from the surroundings |
| $LW_{up}$ | Black body emission |





**Table A2.** Variables describing physical properties of the ground, air and local environment

| Notation | Meaning |
| --- | --- |
| $\alpha_{asp}$ | Asphalt albedo |
| $\alpha_s$ | Surface albedo |
| $\alpha_{snow}$ | Snow albedo |
| $\alpha_{surf}$ | Albedo of the surrounding environment |
| $SVF$ | Sky view factor |
| $\epsilon$ | Emissivity factor |
| $c_a$ | Specific heat capacity of air |
| $c_g$ | Specific heat capacity of ground |
| $\rho_a$ | Density of air |
| $\rho_g$ | Density of ground |
| $\rho_w$ | Density of water |
| $\rho_m$ | Density of moist air |
| $L_{wat}$ | Latent heat of water vaporization |
| $L_{melt}$ | Latent heat for melting |
| $d$ | Zero plane displacement height |
| $z_h$ | Surface roughness parameter for heat |
| $z_m$ | Surface roughness parameter for momentum |
| $K$ | Ground heat conductivity |
| $\rho_g c_g$ | Volumetric heat capacity of the ground |
| $\rho_s c_s$ | Volumetric heat capacity of dry ground |
| $\rho_w c_w$ | Volumetric heat capacity of water |
| $\phi$ | Porosity of the ground |
| $\rho_w$ | Water density |
| $c_w$ | Specific heat capacity of water |
| $m_c$ | Clay fraction of the ground |
| $\gamma$ | Psychrometric constant |





**Table A3.** Variables describing the current state of the ground, surface and atmosphere

| Notation | Meaning |
| --- | --- |
| $T_s$ | Surface temperature |
| $T$ | Ground layer temperature |
| $T_w$ | Water temperature |
| $RH$ | Relative humidity |
| $BLC$ | Boundary layer conductance |
| $u^*$ | Friction velocity |
| $\Psi_h$ | Stability correction factor for heat |
| $\Psi_m$ | Stability correction factor for momentum |
| $\Psi$ | Stability correction factor |
| $p$ | Air pressure |
| $\zeta$ | Stability parameter |
| $e_s$ | Vapour pressure at the surface |
| $e_a$ | Vapour pressure of the air |
| $r_o$ | Aerodynamic resistance |
| $\theta$ | Volumetric water content |

**Table A4.** Physical constants

| Notation | Meaning |
| --- | --- |
| $\sigma_{SB}$ | Stefan-Boltzmann constant |
| $k$ | Von Karman's constant |
| $R_d$ | Gas constant for dry air |
| $g$ | Gravitational constant |





**Table A5.** Variables and parameters related to storage term calculation

| Notation | Meaning |
|---|---|
| $P_{rain}$ | Variable used to determine precipitation type |
| $P_{exp}$ | Variable used to determine precipitation type |
| $St_x$ | Storage x |
| $St_{deposit}$ | Deposit storage |
| $St_{ice}$ | Ice storage |
| $St_{ice2}$ | Secondary ice storage |
| $St_{sum,ice}$ | Total ice content |
| $St_{snow}$ | Snow storage |
| $St_{water}$ | Water storage |
| $St_{water,surf}$ | Water storage on road surface |
| $WS_{rat}$ | Water to snow ratio |
| $Wear_x$ | Reduced amount of storage x in one time step |
| $Wf_x$ | Wearing factor for storage x |
| $Wf_{snow}$ | Wearing factor for snow |
| $Wf_{ice}$ | Wearing factor for ice |
| $Wf_{ice2}$ | Wearing factor for secondary ice storage |
| $Wf_{deposit}$ | Wearing factor for deposit |
| $Wf_{water}$ | Wearing factor for water |
| $E_{m^3}$ | Amount of energy needed to evaporate one cubic meter of water |
| $Q_{melt}$ | Energy that is available for melting |
| $Q_{all}$ | Energy required to melt the full ice/snow layer |
| $EV_w$ | Evaporated/condensed water in time step |
| $Me_{snow}$ | Melted amount of snow in time step |
| $T_{melt}$ | Melting temperature |





**Table A6.** Variables related to time and space and miscellaneous variables.

| Notation | Meaning |
| --- | --- |
| $z$ | Vertical distance in the ground |
| $z_T$ | Air temperature measurement height |
| $z_W$ | Wind measurement height |
| $h$ | Measurement height |
| $Z_{add}$ | Constant used in layer height initialization |
| $t$ | Time |
| $\Delta t$ | Simulation time step |
| $A, B, C, E$ | Coefficients for heat conductivity calculation |
| $A_y$ | Amplitude of deep ground temperature variation during the year |
| $\Omega$ | Frequency of the deep ground temperature variation |
| $J$ | Julian day |
| $Tc$ | Deep round climatological temperature average |
| $Di$ | Displacement of the deep ground temperature curve |
| $m$ | Number of the ground layers |
| $d_a$ | damping depth |
| $C_f$ | Coupling correction coefficient at time t |
| $C_R$ | Coupling correction coefficient |
| $t_c$ | Scaling constant for coupling correction coefficient |
| $X(t)$ | Modified value at time t after relaxation is applied |
| $X_{FO}$ | Forecasted value at the time of the last observation |
| $X_O$ | Last observed value |





*Author contributions.* Virve Karsisto ran the simulations, analyzed the results and wrote the manuscript. She also did the main work in creating the RoadSurf library from the FMI road weather model code.

*Competing interests.* The authors declare that they have no conflict of interest.

*Acknowledgements.* The support provided by the following projects is gratefully acknowledged: SafeTrucks, funded by Business Finland
and the EU EUREKA Xecs program, Winter Premiumin, funded by the European Regional Development Fund, Lapin 5G-kiihdyttämö (Lapland's 5G-accelerator), funded by the European Regional Development Fund, 5G-Safe-Plus, which is part of the Eureka Cluster Celtic-NEXT initiative, funded in Finland by Business Finland.

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
