# Peer review of "RoadSurf 1.1: open-source road weather model library"

_Geoscientific Model Development, 2023_

## Referee Comment (RC1)

This is a nice paper and it serves as a valuable complement to the newly released, freely available source code for the RoadSurf model. The different components of the model are well described. The evaluation part shows that the model is capable of computing the surface temperature well. However, the central theme of the paper is not the model's forecasting capabilities for road surface temperature. Rather it is, like Crevier & Delage (2001), a description of a sophisticated road weather model. As such, the paper should focus its discussion on the model and how it differs from other similar models.

Below, you'll find a few comments and requests for clarification:

The paper would benefit from having a Discussion section.

Abstract: "well suited for forecasting road surface temperature."
The model has a sophisticated storage module that takes asphalt porosity as well as ice, black ice and snow into account. This, among other things, sets it apart from the METRo model. If well implemented, the RoadSurf model should also be well-suited for calculating road conditions, potentially more accurately than METRo. This could be interesting to address in a Discussion section.

Page 4 row 17: "The upward radiation". Explain how the upward radiation from the surroundings affects the road surface.

Page 10 row 23: Explain which items in table 1 refer to which storage term/wear factor x. Is ice2=black ice? How is deposit different from black ice?

Page 11 row 8: What is the disadvantage of allowing water to freeze immediately without affecting the temperature?

Page 12 row 33-34: If the air is dry, can the dew point reach -50 in the north of Finland? "lower than -50 ∘C were removed from the air temperature, surface temperature and dew point temperature"

Page 12 row 46: Why not simply remove those forecasts? They would be of poorer quality than when the system is functioning as expected.

Page 16 row 1: "there was". Otherwise one might misinterpret it as though there is always a decreasing temperature trend in October and January (why not November and December?) in Finland, but the data only supports this for the specific winter season. "there is a general decreasing trend in temperature during those months."

Page 16 row 8-9: This seems counterintuitive. Please explain why! "The 00, 06 and 18 UTC forecast seem to have smaller RMSE values when the actual forecast time is around 12 UTC"

And some minor spelling/grammar errors:

Page 2 row 33: Two is misspelled as "Tow".

Page 3 row 13: "long waver radiation" should be long wave radiation.

Page 3 row 14-15: the forecast point "receive" direct solar radiation. (Karsisto and Horttanainen, 2023)"." should be receives and no period "." after radiation.

Page 4 row 13: This sentence should be rephrased: "The sun position calculation is based on book by Jean Meeus Meeus (1991)."

Page 4 row 27: "Both asphalt albedo snow albedo". Should include an "and".

Page 4 row 29: Albedo is misspelled as "alpedo".
Page 4 row 30: Albedo and calculated are misspelled as " alpedo is calucalted".

Page 5 row 16: The formatting of the citation "calculated with equation Campbell (1985):" should be like on row 23 "(Campbell, 1985)".

Page 7 row 10: Formatting of citation "equation Patankar (1980):"
Page 7 row 15: Formatting of citation "equation Campbell (1985):"

Page 7 row 26: "calculated as as a weighted", two "as".

Page 8 row 22: "divided to 16 " should be "divided into 16 ".

Page 10 row 30: Add "each"( time step) to the sentence "The amount of evaporated or condensed water in time step is calculated with equation:"

Page 11 row 7: Consider using "the entire" instead of "whole".

Page 11 row 9: Consider using "The entire" instead of "Whole".

Page 11 row 16: Add "the (temperature)" to "After calculating the available energy, temperature".

Page 11 row 23: "left over" is one word "leftover".

Page 11 row 24: Add "The (temperature)" to "Temperature for the uppermost layer".

Page 11 row 34: Rephrase this sentence " If water to snow ratio is higher than 0.1 and surface temperature goes below freezing limit, both water storage and snow storage
and transferred to ice storage."

Page 12 row 23: Why 2? "stations in Finland 2."

Page 14 row 3-4: Should be "above (zero degrees)" in "the simulated temperature rises below zero degrees".

Page 14 row 9: Bias is in figure 5 and RMSE in 4, but Bias is mentioned first in the text. "Bias (forecast-observation) and root mean square error (RMSE)"

Page 14 row 10: Add "to" in "reffered to as"

Page 14 row 13: Should be forecasts in "always forecast the".

Page 15 row 3: "Each panel shows results" instead of "Each panel show results".

Page 16 row 8: "forecasts (seem)" in "The 00, 06 and 18 UTC forecast seem".

Page 21 Table A5: "one (time step)" "Evaporated/condensed water in time step"

Page 21 Table A5: "one (time step)" "Melted amount of snow in time step"

Page 22 Table A6: "(Deep) ground" "Deep round climatological temperature average"

---

## Referee Comment (RC3)

==========================================================================
**Specific comments and technical corrections**
==========================================================================

**Section 1 - INTRODUCTION**
PAGE 1, LINE 23: Clarify: "where the temperature drops" vs. "where the air temperature drops"
P1, L33: Clarify: "to predict road conditions" vs. "to predict road weather conditions"
P2: Add extra paragraph at the end of the Section 1 to explain content/structure of the manuscript

**Section 2 – REQUIREMENTS**
PAGE 2, LINE 14: Clarify: Which Fortran?
P2, L14: "The library is not guaranteed to work with older Fortran versions"; write clearly – do you mean with older version(s) of compiler(s)? (your repository has fortran95 files), or additional programs (by potential users) should be written in f95 or higher?
P2, L14: Clarify: "following variables" vs. "following input variables"?
P2, L17: Clarify: "or Humidity (%)" – "or" vs "and" vs "or/and" ? & "Humidity" vs "Relative Humidity"?
P2, L19: Clarify: "Precipitation (mmh−1)" – do you mean "Intensity of Precipitation" in units mm per hour; use space between "mm" and "h-1"
P2, L22: Clarify: "precipitation phase" – include/list in brackets the phases;
P2, L22: Declare "sky view factors" as "SVF", and use acronym afterwards;
P2, L24: Clarify: "whole hemisphere" vs. "whole sky hemisphere "
P2, L27: Clarify: "atmospheric variables should" vs "atmospheric variables listed above should";
P2, L29-31: Clarify: "ground temperature" – "temperature" – "road surface temperature" – "surface temperature"; confusing; write more clearly
P2, L32: Clarify: "Library contains only function" – clarify; GitLab contains not only functions/ programs; write in brief summary on what is stored in repository;
P2, L33: Clarify: "Tow examples" vs. "Two examples"?

**Section 3 - ROADSURF PHYSICS**
PAGE 2, LINE 37-38:  All variables given in Appendix A should have units (include 3rd Column for "Units")
P2, L40: Clarify: "in the model" vs "in the RoadSurf model"
PAGE 3, FIGURE 1: captain to figure 1 should be self-explanatory; all symbols used/plotted on figure should be explained; each flux given below "where" should also include units;
P3, L2: Clarify in "where"-blocks (everywhere throughout the text): Do you mean sign "=" means the hyphen , replace to "—" & replace in all other appropriate places throughput the manuscript;
P3, L4: Clarify: what do you mean "of the other fluxes"; re-write accordingly;
**Section "3.1.1 Net radiation"**
P3, L10: See comment for P3, L2 & for listed variable in "where" - should also include units;
P3, L13: Clarify: "some long waver radiation" vs "some long-wave" radiation;
P3, L13: Unify throughout the text: "long wave" vs "long-wave" & "short wave" vs "short-wave";
PAGE 4, equations on P4, included – "mod" subscript as well as other should be clearly defined in text; define these
In equations (do you use MathEditor?): sign "*" should be replaced by correct multiplication sign "·" in MathEditor & such sign should be added where it is appropriate
P4, L10: Clarify/re-write: "albedo of the surrounding environment" (do you mean albedo at exact point on road? or averaged albedo over mentioned later"  vs. "surrounding vegetation, terrain and buildings"
P4, L13: Provide specific reference & re-phrase from "is based on book by Jean Meeus Meeus (1991)"
**Section "3.1.2 Albedo"**
P4, L27: Replace: "Both asphalt albedo snow albedo are" -> "Both asphalt albedo and snow albedo are"
P4, L29-30: Replace: "alpedo is" -> "albedo is" & "alpedo is calucalted" -> "albedo is calculated"; do you mean "surface albedo" (as albedo in Eq. 5 vs Eq. 10)?
**Section "3.1.3 Sensible heat flux"**
P5, EQ12: Clarify: $\rho a$ = density of air; is it of dry air?
P5, L14: Write correctly units for Rd (gas constant for dry air)
P5 & OTHER PAGES: There is frequent reference to textbooks of Campbell 1985 & 1986; it could be more clearly (easily understandable) written in the text of the manuscript, when there is also, at least, a reference included to Chapters of these textbooks.

P5-P6 (Equations 16-17-18-19): there are for unstable and stable conditions; what about expression/value for neutral conditions?

**Section "3.1.4 Latent Heat Flux"**

P6, L16: Clarify: "LE is set to zero if there is no water to evaporate and LE is positive."; confusing; re-write

**Section "3.2 Heat flow in the ground"**

P7, EQ27: clarify: "T = layer temperature"; is it soil layer temperature? & ρg = density; it is density of soil? or density of soil layer? all variables/constants in all equations should have clear definitions in "where"

**Section "3.2.1 Volumetric heat capacity and heat conductivity"**

P8, L5: "Water density and specific heat capacity have constant values below zero degrees" – clarify, re-write & check units for cw (in kilo-Joules?)

P8, EQ35: check; re-type correctly

The word "temperature" is frequently used in different parts of the manuscript, but in different contents; and in many places it sounds confusing or misleading; please, be more specific and clearly write to which exactly temperature you are referring in different sections of the manuscript.

**Section "3.2.2 Temperature field and layer height initialization"**

P8, L25-26: confusing; clarify; re-write - "The temperature of the layers 1-4 is set to the observed temperature (which temperature?) at the initialization. If observations are not available, the temperature of the (soil?) layers is set to the air temperature (air? or surface temperature?)."

**Section "3.3 Coupling"**

P9, Table 1: this Table included in section 3.3, but the first time referenced in section 3.5; update and place correctly; some listed events do not have signs (+ or -); clarify; write self-explanatory caption text to Table 1

P9, L8-9: "the simulation goes back three hours"; do you mean the model re-run? clarify, re-write

P9, L11: "until the simulated surface temperature is within 0.1 ◦C from the observed temperature"; do you mean +/- 0.1 deg? clarify; re-write

**Section "3.5.1 Precipitation"**

P10, L11-12: in written explanations, to which type exactly the values 0.3 and 0.7 are corresponded; use not only less or more math.signs

**Section "3.5.2 Wear"**

P10, EQ43: explain $\Delta t$ and its units;

**Section "3.6 Freezing and melting"**

P11, EQ47 all included variables should be declared below as "where"; not all previously has been used

**Section - 4 DATA**

**Section "4.1 Forecast"**

P12, L9: Clarify: "it is updated once an hour"; re-write

P12, L11: "different NWP models (ECMWF, MEPS, GFS)" – mixture of organizations and models; do you mean IFS, Harmonie, GFS models? re-write

P12, L19-20: "Because the data are in grid format, the forecast data were interpolated to the road weather station points"; do you mean: grid (or mistyped GRIB?) format - in gridded format of latitude-longitude coordinates? points – geographical coordinates? clarify; re-write

**Section "4.2 Road weather observations"**

P12, L23: Clarify: what does mean "in Finland 2"?; re-write

P12, L25: Clarify: which humidity (relative, specific?); which precipitation (type, intensity, amount?)

P12, L33: "Third, the values that were greater than 50 ◦C" – can the road surface temperature (for example, for a new asphalt on a road) be more than 50 deg in hot summer days? and how many % of such days/measurements were removed from time series of observations? Or the RoadSurf is only applicable for forecasting of the winter road weather conditions (if yes, write it clearly somewhere in the Introduction-section)

**Section "4.3 Simulation data structure"**

P12, L37: Clarify: "To evaluate the performance of the RoadSurf library", do you mean – "To evaluate the performance of the Road Weather Model with newly implemented capabilities of the RoadSurf library"?; re-write

P12, L39: "GitHub repository as "Example 2" in the examples folder …" – add here also direct path as done, for example, in section 4.1

P12, L43: although UTC is well-known, but explain (Universal Coordinated Time), when the first time is used in text

**Section "4.4 Example hindcast"**

P12, L50: Clarify: "Sensor 2 gives somewhat colder measurements than sensor 1." – explain/add in short on which reasons(s) could be.

P13, Figure 2: (i) there is no reference to Figure 2 in the text of the manuscript; should be included where it is needed; (ii) caption text to Figure 2 should be self-explanatory (fx. "Geographical locations of road weather stations (operated by …) in Finland"; (iii) plot/add to Figure names of countries, including Finland; (iv) adding latitude and latitude values would be essential to include.

P13, Figure 3: (i) horizontal axis marks are slightly confusing "DD Month ##" -> add "UTC", or clearly writ in caption text as "was 29th December 2022, 00 UTC"; (ii) mark location of road weather station in Porvoo in Figure 2; (iii) as you have Figure 3a & Figure 3b -> use these instead of "The uppermost panel" & "The lower panel"; (iv) on vertical axis: Fig3a "Tsurf" is "Ts" – road surface temperature? correct symbol, add words as in Fig3b; Fig3b – is symbol "St", add too; (v) omit "The time on x-axis is in UTC time"

**Section 5 - ROADSURF EVALUATION"**

P14, Figure 4: it would be better to include this figure after the Section 5 started, not in the previous section 4.4
The statement "The colors were chosen with the help of colorbrewer2 (https://colorbrewer2.org)" in Figures 4,5, etc. is it necessary to include on each illustration, or just add once to the Acknowledgement-section

P14, L8-9: Update: Following presented sequence of Figures: "Figures 4 and 5 show Bias (forecast-observation) and root mean square error (RMSE) …" -> "Figures 4 and 5 show the root mean square error (RMSE) and Bias (forecast-observation), correspondingly, …" or re-number the Figures (same for Figures 6 and 7)

P16, L6: Clarify: "During those months", do you mean "During these months"? or to which months do you refer?

P14-16, Clarify: in different lines: do you mean a positive bias?; replace "warm bias" -> "positive bias"

P14, L13: Clarify: do you mean a negative bias?; replace "cold bias" -> "negative bias"

P14, L16: May be: "are not considered reliable enough" vs. "are not sufficiently representative"

**Section 6 – CONCLUSIONS**

P16-17: This section needs to be re-written and, at least, included sumup paragraphs about RoadSurf Library description on included components and its content; results of evaluation containing not only "words", but also obtained "numbers" for selected studied period; applicability; plans (if any) for further development/improvement of RoadSurf Library.

**Section "Appendix A"**

PAGES 17-22: All Tables should include additional column containing units for listed variables (Tables 1-3, 5-6) and physical constants (Table 4).

**Section "Competing interests"**

PAGE 23: Re-write as for one person (not as plural); the list of authors includes only one person

**Section "Acknowledgements"**

PAGE 23: As you have run/performed run a lot of simulations, do you acknowledge also using of computing resources, which & where?

**Section "References"**

PAGES 23-24: Note that some references – Crevier & Delage, 2001; Fritsch et al., 2000; Glahn & Lowry, 1972 - are not accessible by doi-links provided (not correctly given?); correct as necessary & double-check the accuracy of provided references in this section.

---

## Author Comment (AC1)

Thank you for your valuable comments and grammar corrections. I appreciate your thorough reading of the manuscript. Attached are point by point replies to the comments. The pointed out spelling/grammar errors will be corrected in the revised manuscript.

This is a nice paper and it serves as a valuable complement to the newly released, freely available source code for the RoadSurf model. The different components of the model are well described. The evaluation part shows that the model is capable of computing the surface temperature well. However, the central theme of the paper is not the model's forecasting capabilities for road surface temperature. Rather it is, like Crevier & Delage (2001), a description of a sophisticated road weather model. As such, the paper should focus its discussion on the model and how it differs from other similar models.

-Thank you for your suggestion. Comparison to other road weather models would indeed be interesting. However, I think that including comparison to other road weather models would broaden the scope too much for this paper and would require an additional paper. There are multitude of road weather models which share the same basic physical principles and thus meaningful comparison would require going to too much detail for a discussion chapter. One possibility is focusing comparison only to the METRo model that is the only other open-source road weather model. However, as Crevier's and Delage's paper is already over a decade old and as the model code is contributed by many parties, some of the details might have changed over time. This would make the comparison difficult. In addition, comparing details like the number of ground layers or how the boundary layer conductance is calculated would not be meaningful for many readers without showing how they affect the model results. This would require running both models and comparing the results, which would broaden the paper too much.

Below, you'll find a few comments and requests for clarification:
The paper would benefit from having a Discussion section.
-May I kindly request some clarification regarding your comment? Given that a comparison with other road weather models might significantly broaden the scope of our manuscript, could you please suggest what type of discussion would be appropriate to include?

Abstract: "well suited for forecasting road surface temperature."
The model has a sophisticated storage module that takes asphalt porosity as well as ice, black ice and snow into account. This, among other things, sets it apart from the METRo model. If well implemented, the RoadSurf model should also be well-suited for calculating road conditions, potentially more accurately than METRo. This could be interesting to address in a Discussion section.
-Both METRo and RoadSurf predict storage terms, but it is true that while METRo has only two storages: water and combined storage for ice and snow, RoadSurf have four different storages: water, snow, ice and black ice. However, determining the accuracy of these predictions is difficult without reliable observations. The optical instruments at the road weather stations measure the thicknesses of water, ice and snow layers, but are not reliable enough for accurate measurements. In addition, these amounts depend greatly on the spot on the road they are measured. Thus, it is difficult to assess which one of the models is more accurate. We would not like to speculate in discussion which of the models is better without actual verification results.

Page 4 row 17: "The upward radiation". Explain how the upward radiation from the surroundings affects the road surface.

-An more detailed explanation will be added to the revised manuscript "Smaller sky view factor decreases the amount of long wave radiation from the sky, but increases the long wave radiation from surroundings. As approximation of the radiation from the surroundings, the model uses upward radiation from an NWP model. As the radiation output from an NWP model represents the whole grid cell, the upward long wave radiation can be assumed to present the road surroundings rather than the road that covers only small part of the grid. Although upward radiation is not same as radiation towards the road point, it can be used as rough approximation. It is calculated as the difference between the net long wave radiation and incoming long wave radiation:"

Page 10 row 23: Explain which items in table 1 refer to which storage term/wear factor x. Is ice2=black ice? How is deposit different from black ice?
-Simple explanation will be added below the wear factors. Deposit and secondary ice storage are explained earlier, so they are not elaborated further. (Page 9 row 26: "Deposit means black ice that has formed on the road surface via deposition." Page 9 row 28: "There are two separate storages for ice that are otherwise similar but the secondary ice storage is reduced faster by traffic. ")

Page 11 row 8: What is the disadvantage of allowing water to freeze immediately without affecting the temperature?
-In the real world, water gradually freezes and freezing releases heat. Thus, letting the water freeze immediately may cause too cold surface temperature and too fast ice formation. However, the phenomena's accurate simulation is difficult as it would require increasing surface temperature when freezing, which would cause temperature to rise above the freezing limit. This minght cause surface temperature to bounce above and below freezing limit when water is freezing.

Page 12 row 33-34: If the air is dry, can the dew point reach -50 in the north of Finland? "lower than -50 ∘C were removed from the air temperature, surface temperature and dew point temperature"
-It can, but it is rare, so it probably does not affect much to the results

Page 12 row 46: Why not simply remove those forecasts? They would be of poorer quality than when the system is functioning as expected.
-This is a good suggestion; we will remove those forecasts and recalculate the results for the revised manuscript

Page 16 row 1: "there was". Otherwise one might misinterpret it as though there is always a decreasing temperature trend in October and January (why not November and December?) in Finland, but the data only supports this for the specific winter season. "there is a general decreasing trend in temperature during those months."
-This will be corrected to the revised manuscript

Page 16 row 8-9: This seems counterintuitive. Please explain why! "The 00, 06 and 18 UTC forecast seem to have smaller RMSE values when the actual forecast time is around 12 UTC"
-The reason for this will be investigated more closely and if explanation is found it will be added to the revised manuscript

---

## Author Comment (AC3)

*General comments*

*The manuscript entitled "RoadSurf 1.1: open-source road weather model library" is a good and important contribution to scientific modelling community as well as to potential end-users from road authorities/directorates and public in many countries. Providing open public access to repository (a good way to go) is crucial and important step for enlarging community of users and developers of the RoadSurf road weather model (RWM). The chosen approach and applied methods are valid as well as results presented and discussed in appropriate and balanced way in the manuscript. The given description (incl. repository) of the RoadSurf 1.1 RWM library is sufficient to reproducibility and traceability of results.*

*With respect to presentation quality, it would be useful to add an extra paragraph at end of Section 1 to explain content/structure of the manuscript. The section "Conclusions" needs to be re-written and, at least, included sumup paragraphs about RoadSurf Library description on included components and its content; results of evaluation containing not only "words", but also obtained "numbers" for selected studied period; applicability; plans (if any) for further development/improvement of RoadSurf Library. The illustrations should have a self-explanatory written text in captions to Figures and Tables (see attached file with detailed comments/remarks/suggestions/etc). All Tables in Annex should include additional column containing units for listed variables (Tables 1-3, 5-6) and physical constants (Table 4), or may be? include at the beginning of the manuscript as "Nomenclature". Frequently, the word "temperature" is used through the text of the manuscript, but it has different "attribution" as: air temperature, road surface temperature, soil temperature, dew point temperature, etc. – these should re- checked and accurately written to avoid a confusion or misleading. It seems that accurate spell-checking of the manuscript text is needed (before submission) to exclude accident mistyping.*

*Specific comments and Technical corrections are included in the attached file.*

Thank you for your valuable feedback on the manuscript. I appreciate your thorough review and have taken your comments into consideration in the revised version of the manuscript. Here are answers to the general comments. Point by point replies to the specific comments can be found in the attached file.

-Paragraph explaining manuscript structure was added
-A concise summary of the most crucial features of the RoadSurf Library was added. This should provide readers with a clear understanding of its capabilities.
- I believe that listing all components in the conclusions section might be excessive. Instead, main features were highlighted.
-In the context of the conclusions, I have opted for a qualitative approach than writing the precise numbers. The focus is on conveying the overall impact and applicability of the library rather than presenting specific numerical values.
-Applicability: Explanation about library's usage was added to the conclusions
-A mention about plans to convert the library to Python was added to the conclusions.
-Figures, Tables and captions were improved based on the specific comments
-Units will be added to the appendix tables and to "where" sections of the manuscript
-Clarifications to the usage of "temperature" term were done where appropriate
-The manuscript will go through Copernicus's English language copy editing service before publication, which will ensure that the manuscript adheres to a high standard of grammar.

Once again, thank you for your thoughtful feedback. I hope these revisions enhance the clarity of the manuscript.

========================================================================
**Specific comments and technical corrections**
========================================================================
**Section 1 - INTRODUCTION**
PAGE 1, LINE 23: Clarify: "where the temperature drops" vs. "where the air temperature drops"
-Added "surface" before temperature

P1, L33: Clarify: "to predict road conditions" vs. "to predict road weather conditions"
-Changed to "road weather conditions"

P2: Add extra paragraph at the end of the Section 1 to explain content/structure of the manuscript
-Added

**Section 2 – REQUIREMENTS**
PAGE 2, LINE 14: Clarify: Which Fortran?
-version (Fortran 2008) was added

P2, L14: "The library is not guaranteed to work with older Fortran versions"; write clearly – do you mean with older version(s) of compiler(s)? (your repository has fortran95 files), or additional programs (by potential users) should be written in f95 or higher?
-chaged from "Fortran versions" to "Fortran compilers"

P2, L14: Clarify: "following variables" vs. "following input variables"?
-Added "input" before variables

P2, L17: Clarify: "or Humidity (%)" – "or" vs "and" vs "or/and" ? & "Humidity" vs "Relative Humidity"?
-"or" is correct here. Added "Relative" before humidity

P2, L19: Clarify: "Precipitation (mmh−1)" – do you mean "Intensity of Precipitation" in units mm per hour; use space between "mm" and "h-1"
-Added "intensity" after precipitation and space between units

P2, L22: Clarify: "precipitation phase" – include/list in brackets the phases;
-Added explanation in brackets

P2, L22: Declare "sky view factors" as "SVF", and use acronym afterwards;
-Added SVF and changed sky view factor -> SVF afterwards

P2, L24: Clarify: "whole hemisphere" vs. "whole sky hemisphere "
-Added "sky" before hemisphere

P2, L27: Clarify: "atmospheric variables should" vs "atmospheric variables listed above should";
-added "listed above"

P2, L29-31: Clarify: "ground temperature" – "temperature" – "road surface temperature" – "surface temperature"; confusing; write more clearly
-added "ground" before temperature profile

P2, L32: Clarify: "Library contains only function" – clarify; GitLab contains not only functions/ programs; write in brief summary on what is stored in repository;
-GitHub is used to store the library with an example program that implements library functions. It was added to the text that repository contains also example datasets and user manual.

P2, L33: Clarify: "Tow examples" vs. "Two examples"?
-Changed "Tow" -> "Two"

**Section 3 - ROADSURF PHYSICS**
PAGE 2, LINE 37-38: All variables given in Appendix A should have units (include 3rd Column for "Units")
-Units will be added to the revised version of the manuscript

P2, L40: Clarify: "in the model" vs "in the RoadSurf model"
-Changed to "in the RoadSurf library"

PAGE 3, FIGURE 1: captain to figure 1 should be self-explanatory; all symbols used/plotted on figure should be explained;
-Added explanations for variables

each flux given below "where" should also include units;
-These will be added to the revised version of the manuscript

P3, L2: Clarify in "where"-blocks (everywhere throughout the text): Do you mean sign "=" means the hyphen , replace to "—" & replace in all other appropriate places throughput the manuscript;
-"=" is used to indicate that G is ground heat flux, for example

P3, L4: Clarify: what do you mean "of the other fluxes"; re-write accordingly;
-added list of other fluxes

**Section "3.1.1 Net radiation"**
P3, L10: See comment for P3, L2 & for listed variable in "where" - should also include units;
-"=" is appropriate here

P3, L13: Clarify: "some long waver radiation" vs "some long-wave" radiation;
-Changed to "long-wave"

P3, L13: Unify throughout the text: "long wave" vs "long-wave" & "short wave" vs "short-wave";
-changed to "long-wave" and "short-wave"

PAGE 4, equations on P4, included – "mod" subscript as well as other should be clearly defined in text; define these
-Added explanation for $SW_{dif, mod}$

In equations (do you use MathEditor?): sign "*" should be replaced by correct multiplication sign "·" in MathEditor & such sign should be added where it is appropriate
-Changed throughout the text

P4, L10: Clarify/re-write: "albedo of the surrounding environment" (do you mean albedo at exact point on road? or averaged albedo over mentioned later" vs. "surrounding vegetation, terrain and buildings"
-Added "average"

P4, L13: Provide specific reference & re-phrase from "is based on book by Jean Meeus Meeus (1991)"
-Rewrote sentence "The calculation of the sun's position is based on the methods presented in the book by Jean Meeus (Meeus, 1991)"

**Section "3.1.2 Albedo"**
P4, L27: Replace: "Both asphalt albedo snow albedo are" -> "Both asphalt albedo and snow albedo are"
-replaced

P4, L29-30: Replace: "alpedo is" -> "albedo is" & "alpedo is calucalted" -> "albedo is calculated"; do you mean "surface albedo" (as albedo in Eq. 5 vs Eq. 10)?
-Changed, added "surface for clarification"

**Section "3.1.3 Sensible heat flux"**
P5, EQ12: Clarify: $\rho_a$ = density of air; is it of dry air?
-"added " dry for  clarification

P5, L14: Write correctly units for Rd (gas constant for dry air)
-Changed "Kg" to "kg"

P5 & OTHER PAGES: There is frequent reference to textbooks of Campbell 1985 & 1986; it could be more clearly (easily understandable) written in the text of the manuscript, when there is also, at least, a reference included to Chapters of these textbooks.
-While specific chapter references could enhance clarity, we believe that the current citations provide sufficient context for readers to locate the relevant material.

P5-P6 (Equations 16-17-18-19): there are for unstable and stable conditions; what about expression/value for neutral conditions?
-Added that equations 18 and 19 are also used in neutral conditions.

**Section "3.1.4 Latent Heat Flux"**
P6, L16: Clarify: "LE is set to zero if there is no water to evaporate and LE is positive."; confusing; re-write
-This was moved to the end of the section with explanation.

**Section "3.2 Heat flow in the ground"**
P7, EQ27: clarify: "T = layer temperature"; is it soil layer temperature?
-Added "ground" before layer

 & $\rho_g$ = density; it is density of soil? or density of soil layer? all variables/constants in all equations should have clear definitions in "where"
-Changed to "density of the ground"

**Section "3.2.1 Volumetric heat capacity and heat conductivity"**
P8, L5: "Water density and specific heat capacity have constant values below zero degrees" – clarify, re-write & check units for cw (in kilo-Joules?)
-Re-wrote as: "Below zero degrees, $\rho_w$ and $c_w$ are get values of ice density and specific heat capacity of ice". Removed "k" before J

P8, EQ35: check; re-type correctly
-Corrected square root and removed brackets from mc

The word "temperature" is frequently used in different parts of the manuscript, but in different contents; and in many places it sounds confusing or misleading; please, be more specific and clearly write to which exactly temperature you are referring in different sections of the manuscript.
-Additions were made throughout the manuscript

**Section "3.2.2 Temperature field and layer height initialization"**
P8, L25-26: confusing; clarify; re-write - "The temperature of the layers 1-4 is set to the observed temperature (which temperature?) at the initialization. If observations are not available, the temperature of the (soil?) layers is set to the air temperature (air? or surface temperature?)."
-Added clarifications

**Section "3.3 Coupling"**
P9, Table 1: this Table included in section 3.3, but the first time referenced in section 3.5; update and place correctly; some listed events do not have signs (+ or -); clarify; write self-explanatory caption text to Table 1
-Added explanations for +, - and empty space and moved the table

P9, L8-9: "the simulation goes back three hours"; do you mean the model re-run? clarify, re-write
-Lines 10-11 explain that the three-hour period is rerun

P9, L11: "until the simulated surface temperature is within 0.1 ∘C from the observed temperature"; do you mean +/- 0.1 deg? clarify; re-write
-added "+/-" before 0.1 C

**Section "3.5.1 Precipitation"**
P10, L11-12: in written explanations, to which type exactly the values 0.3 and 0.7 are corresponded; use not only less or more math.signs
-Changed signs for sleet categorization

**Section "3.5.2 Wear"**
P10, EQ43: explain Δt and its units;
-Added explanation for Δt

**Section "3.6 Freezing and melting"**
P11, EQ47 all included variables should be declared below as "where"; not all previously has been used
-Added missing variables, changed T_s to T_1 for clarity

**Section - 4 DATA**
**Section "4.1 Forecast"**
P12, L9: Clarify: "it is updated once an hour"; re-write
-Changed "updated" to "run"

P12, L11: "different NWP models (ECMWF, MEPS, GFS)" – mixture of organizations and models; do you mean IFS, Harmonie, GFS models? re-write
-Changed to "ECMWF's high resolution forecast, MetCoOp Ensemble Prediction System's (MEPS) control member, National Centers for Environmental Prediction (NCEP) Global Forecast System (GFS) forecast"

P12, L19-20: "Because the data are in grid format, the forecast data were interpolated to the road weather station points"; do you mean: grid (or mistyped GRIB?) format - in gridded format of latitude-longitude coordinates? points – geographical coordinates? clarify; re-write
-Changed from "grid format" to "gridded format"

**Section "4.2 Road weather observations"**
P12, L23: Clarify: what does mean "in Finland 2"?; re-write
-Added "Figure" before 2

P12, L25: Clarify: which humidity (relative, specific?); which precipitation (type, intensity, amount?)
-Changed to "relative humidity" and "precipitation intensity"

P12, L33: "Third, the values that were greater than 50 ◦C" – can the road surface temperature (for example, for a new asphalt on a road) be more than 50 deg in hot summer days? and how many % of such days/measurements were removed from time series of observations? Or the RoadSurf is only applicable for forecasting of the winter road weather conditions (if yes, write it clearly somewhere in the Introduction-section)
-The asphalt temperature can go above 50 C in summer, but the study included only months from October to March, so no actual measurements were removed. It has been stated at the introduction (P2, L11-12) that the model is not optimized to predict very high temperatures.

**Section "4.3 Simulation data structure"**
P12, L37: Clarify: "To evaluate the performance of the RoadSurf library", do you mean – "To evaluate the performance of the Road Weather Model with newly implemented capabilities of the RoadSurf library"?; re-write
-As the manuscript focuses on RoadSurf library, the meaning is to evaluate the performance of the library as was written

P12, L39: "GitHub repository as "Example 2" in the examples folder …" – add here also direct path as done, for example, in section 4.1
-Added link

P12, L43: although UTC is well-known, but explain (Universal Coordinated Time), when the first time is used in text
-Added explanation

**Section "4.4 Example hindcasts**
P12, L50: Clarify: "Sensor 2 gives somewhat colder measurements than sensor 1." – explain/add in short on which reasons(s) could be.
-Added "Reason for this can be that sensor 2 is installed on a colder location on the road or that one of the sensors is calibrated incorrectly."

P13, Figure 2: (i) there is no reference to Figure 2 in the text of the manuscript; should be included where it is needed;
-Added at the beginning of section 4.2

(ii) caption text to Figure 2 should be self-explanatory (fx. "Geographical locations of road weather stations (operated by …) in Finland";
-Changed to "Geographical locations of road weather stations in Finland operated by Intelligent Traffic Management Finland."

 (iii) plot/add to Figure names of countries, including Finland;
-As the caption now states that the map shows Finland, I don't think that adding country names is necessary.

 (iv) adding latitude and latitude values would be essential to include.

-Latitudes and longitudes were added

P13, Figure 3: (i) horizontal axis marks are slightly confusing "DD Month ##" -> add "UTC", or clearly writ in caption text as "was 29th December 2022, 00 UTC";
-Added UTC

(ii) mark location of road weather station in Porvoo in Figure 2;
-I don't think it would add much value and could even add confusion, as the location is a mere example and does not have any significant value. Coordinates were added to the caption instead

(iii) as you have Figure 3a & Figure 3b -> use these instead of "The uppermost panel" & "The lower panel";
-Changed

(iv) on vertical axis: Fig3a "Tsurf" is "Ts" – road surface temperature? correct symbol, add words as in Fig3b; Fig3b – is symbol "St", add too;
-Changed "Tsurf" to "T_surface" and added explanation to caption. I think that "Storage" as the axis label is better as it can be directly understood

 (v) omit "The time on x-axis is in UTC time"
-Removed

**Section 5 - ROADSURF EVALUATION"**
P14, Figure 4: it would be better to include this figure after the Section 5 started, not in the previous section 4.4
-Moved

The statement "The colors were chosen with the help of colorbrewer2 (https://colorbrewer2.org)" in Figures 4,5, etc. is it necessary to include on each illustration, or just add once to the Acknowledgement-section
-Moved to acknowledgments

P14, L8-9: Update: Following presented sequence of Figures: "Figures 4 and 5 show Bias (forecast-observation) and root mean square error (RMSE) …" -> "Figures 4 and 5 show the root mean square error (RMSE) and Bias (forecast-observation), correspondingly, …" or re-number the Figures (same for Figures 6 and 7)
-Switched order

P16, L6: Clarify: "During those months", do you mean "During these months"? or to which months do you refer?
-Changed "during those months" to "during October and January"

P14-16, Clarify: in different lines: do you mean a positive bias?; replace "warm bias" -> "positive bias"
-Changed from "warm" to "positive"

P14, L13: Clarify: do you mean a negative bias?; replace "cold bias" -> "negative bias"
-Changed from "cold" to "negative"

P14, L16: May be: "are not considered reliable enough" vs. "are not sufficiently representative"
-I think that reliable is better term here, as the question is also about sensor reliability in addition how representative it is

**Section 6 – CONCLUSIONS**
P16-17: This section needs to be re-written and, at least, included sumup paragraphs about RoadSurf Library description on included components and its content; results of evaluation containing not only "words", but also obtained "numbers" for selected studied period; applicability; plans (if any) for further development/improvement of RoadSurf Library.
 -Please see answers to the general comments on the manuscript at the beginning of the file

**Section "Appendix A"**
PAGES 17-22: All Tables should include additional column containing units for listed variables (Tables 1-3, 5-6) and physical constants (Table 4).
-Units will be added to the revised version of the manuscript

**Section "Competing interests"**
PAGE 23: Re-write as for one person (not as plural); the list of authors includes only one person
-changed to singular

**Section "Acknowledgements"**
PAGE 23: As you have run/performed run a lot of simulations, do you acknowledge also using of computing resources, which & where?
-The computing was performed on FMI's own facilities

**Section "References"**
PAGES 23-24: Note that some references – Crevier & Delage, 2001; Fritsch et al., 2000; Glahn & Lowry, 1972 - are not accessible by doi-links provided (not correctly given?); correct as necessary & double-check the accuracy of provided references in this section.
-Links will be checked and fixed for the revised version of the manuscript

---

## Author Response (AR2)

Editors comment:

*Please add more discussion on the reviewer #1 to the manuscript.*
*One example is the reviewer#1's comment on Page 11 row 8. Your replies on some difficulties to simulate freezing process will be useful for readers of your paper.*

Thank you for you suggestion. Following additions were made to the manuscript:

Discussion about freezing process was added to section 3.6: "This may cause too cold surface temperature and too fast ice formation in the simulation. However, the phenomena s accurate modeling is difficult as it would require increasing surface temperature when freezing, which would cause temperature to rise above the freezing limit. This might cause surface temperature to bounce above and below freezing limit during freezing."

An addition was made to section 4.2 regarding data quality checking process: "This might have caused removal of some real data, but it is not expected affect considerably to the results."

---

## Author Response (AR3)

*Dear Virve Karsisto*

*I am sorry for my late response. I checked your GitHub link and I realized that its address is broken. That is, link address should be "https://github.com/fmidev/RoadSurf.", not "https://github.com/fmidev/RoadSurf/)."*

*Also, please use zenodo for the fixed codes used in your study instead of GitHub.*

*Thank you for your understanding.*

*Sincerely,*

*Junky*
* * *
Dear Editor,

Thank you very much for your clarification. I have fixed the Github link. There seems to be a minor misunderstanding, as I have already provided the Zenodo link for the fixed version and the Github link I have given refers only to the newest version of the model. The reference to the Zenodo archive is right at the start of the code and data availability statement: "The RoadSurf code used in this study is available in Zenodo (Karsisto et al., 2023) under MIT licence." The link to Zenodo can be found in the references section:

Karsisto, V., Kangas, M., Heiskanen, M., Hippi, M., Ruotsalainen, J., Heikinheimo, M., and Backman, L.: RoadSurf 1.1 [code], https://doi.org/10.5281/zenodo.8246081, 2023.
In this I have followed the example found in the code and data policy:

https://www.geoscientific-model-development.net/policies/code_and_data_policy.html

*"The current version of model is available from the project website: url under the licence name licence. The exact version of the model used to produce the results used in this paper is archived on Zenodo (citation), as are input data and scripts to run the model and produce the plots for all the simulations presented in this paper (citation)."*

Best regards,

Virve Karsisto